# Allergenicity and Bioavailability of Nickel Nanoparticles Compared to Nickel Microparticles in Mice

**DOI:** 10.3390/ma16051834

**Published:** 2023-02-23

**Authors:** Dai Tsuchida, Yuko Matsuki, Jin Tsuchida, Masahiro Iijima, Maki Tanaka

**Affiliations:** 1Division of Orthodontics and Dentofacial Orthopedics, Department of Oral Growth and Development, School of Dentistry, Health Sciences University of Hokkaido, Hokkaido 061-0293, Japan; 2Department of Clinical Laboratory Science, School of Medical Technology, Health Sciences University of Hokkaido, Hokkaido 061-0293, Japan

**Keywords:** nickel allergy, nickel nanoparticles, nickel allergy model mice

## Abstract

Metal allergy is a common disease that afflicts many people. Nevertheless, the mechanism underlying metal allergy development has not been completely elucidated. Metal nanoparticles might be involved in the development of a metal allergy, but the associated details are unknown. In this study, we evaluated the pharmacokinetics and allergenicity of nickel nanoparticles (Ni-NPs) compared with those of nickel microparticles (Ni-MPs) and nickel ions. After characterizing each particle, the particles were suspended in phosphate-buffered saline and sonicated to prepare a dispersion. We assumed the presence of nickel ions for each particle dispersion and positive control and orally administered nickel chloride to BALB/c mice repeatedly for 28 days. Results showed that compared with those in the Ni-MP administration group (MP group), the Ni-NP administration group (NP group) showed intestinal epithelial tissue damage, elevated serum interleukin (IL)-17 and IL-1β levels, and higher nickel accumulation in the liver and kidney. Additionally, transmission electron microscopy confirmed the accumulation of Ni-NPs in the livers of both the NP and nickel ion administration groups. Furthermore, we intraperitoneally administered a mixed solution of each particle dispersion and lipopolysaccharide to mice and then intradermally administered nickel chloride solution to the auricle after 7 days. Swelling of the auricle was observed in both the NP and MP groups, and an allergic reaction to nickel was induced. Particularly in the NP group, significant lymphocytic infiltration into the auricular tissue was observed, and serum IL-6 and IL-17 levels were increased. The results of this study showed that in mice, Ni-NP accumulation in each tissue was increased after oral administration and toxicity was enhanced, as compared to those with Ni-MPs. Orally administered nickel ions transformed into nanoparticles with a crystalline structure and accumulated in tissues. Furthermore, Ni-NPs and Ni-MPs induced sensitization and nickel allergy reactions in the same manner as that with nickel ions, but Ni-NPs induced stronger sensitization. Additionally, the involvement of Th17 cells was suspected in Ni-NP-induced toxicity and allergic reactions. In conclusion, oral exposure to Ni-NPs results in more serious biotoxicity and accumulation in tissues than Ni-MPs, suggesting that the probability of developing an allergy might increase.

## 1. Introduction

Various types of metals are currently used in clinical dentistry. It has been reported that metal allergies are often caused by dental metals, and this has been perceived as a problem. Metals used in the oral cavity, such as orthodontic appliances and restorative crown materials, are always exposed to a complex oral environment. Accordingly, contact with saliva, oral bacteria, and dissimilar metals lead to the corrosion and ionization of dental metals [1,2,3]. It has been reported that these factors cause metal ions to enter the body and trigger the onset of metal allergies [4,5].

Metal allergy is classified as a type IV allergy and is also referred to as a delayed-type allergy because symptoms appear approximately 24–48 h after contact with the antigen. Representative diseases associated with metal allergies in the dental field include contact dermatitis, which develops at remote sites such as on the hands and feet, and oral lichen planus, which frequently occurs at metal contact sites in the oral cavity [6,7]. The mechanism underlying the onset of a type IV allergy, including metal allergies, can be divided into two phases—sensitization and elicitation. At the time of initial contact with an antigen, the antigen that enters the body is taken up by antigen-presenting cells, such as dendritic cells and macrophages, and naïve T cells are activated in local lymph nodes. Simultaneously, memory T cells, which store antigen information, are generated (sensitization phase). During the second and subsequent infiltration [5,6,7] step, memory T cells are activated, and pro-inflammatory cytokines are released, which causes inflammation (elicitation phase) [8].

The patch test is currently the most widely used method for diagnosing metal allergies [9,10,11]. For this, a reagent that contains each allergen is applied to the skin for a certain amount of time, and the reaction that is observed is used to determine the presence or absence of inflammation based on certain criteria. A global epidemiological survey estimated that approximately 17% of women and approximately 3% of men are allergic to nickel, and this prevalence was thought to be even higher in those with skin disease [12,13,14]. The first-line treatment for metal allergies is to avoid contact with the causative metal. In the case of clinical dentistry, if metal is used for crown restoration, this must be removed, which imposes a temporal, economic, and psychological burden on the patient [15].

Recently, in the dentistry field, nanoparticles have been applied to various materials, such as the addition of titanium dioxide nanoparticles to toothpaste to improve abrasiveness and the addition of nanofiller particles to composite resin for dental crown restoration [16]. Moreover, owing to their large specific surface area, nanoparticles have physical properties that are different from bulk material and are used in various fields, such as engineering and medicine, with associated technology expanding rapidly. However, at the same time, it has been reported that metal nanoparticles can cause allergic reactions [17].

In this study, we focused on nickel, which is associated with a high metal allergy incidence. In dentistry, nickel is contained in materials such as orthodontic appliances to move teeth and stainless steel used for dental reinforcement, as well as in clasps. Accordingly, the possibility that nickel nanoparticles are generated due to the application of these materials to the oral cavity and grinding in the oral cavity using a rotary cutter cannot be rejected. Therefore, in this study, we investigated the pharmacokinetics and allergenicity of nickel nanoparticles to clarify their effects on living organisms.

## 2. Materials and Methods

### 2.1. Preparation of Particle Dispersion

In this study, we used nickel nanoparticles (Ni-NPs; 20 nm Ni > 99.9%, EM Japan, Tokyo, Japan) and nickel microparticles (Ni-MPs; Ni > 99.9%, Sigma Aldrich, St. Louis, MO, USA). Each particle was suspended in phosphate buffered saline (PBS) and sonicated for 1 min using an ultrasonic homogenizer (UD-100; TOMY, Tokyo, Japan) to prepare a dispersion.

### 2.2. Physical Characterization of Particles

#### 2.2.1. Morphological Observation and Elemental Analysis

For each particle used, the particle shape was observed using a transmission electron microscope (JEM2100F; JEOL, Tokyo, Japan), and each particle diameter (primary particle size) was measured, with the average particle size calculated. We conducted surface texture and elemental composition analysis using a field emission scanning electron microscope (JSM-7800F; JEOL) and energy dispersive X-ray spectroscopy.

#### 2.2.2. Calculation of Specific Surface Area

The specific surface area of each particle in the powder was measured using an adsorption measuring device (Autosorb 6AG, Yuasa-ionics, Kanagawa, Japan), and the surface area per unit weight was calculated using the BET method.

#### 2.2.3. Particle Size Distribution Measurement

After preparing each particle dispersion, the particle size distribution (i.e., secondary particle size) was measured with a particle size distribution analyzer (HRA9320-X100; MICRO TRAC, Tokyo, Japan). The particle dispersion was adjusted using ultrapure water owing to the characteristics of the device.

#### 2.2.4. Ion Elution Test

Changes in the nickel ion content after the preparation of each particle dispersion were measured immediately after preparation of the dispersion, 12 h after preparation, and 24 h after preparation. Each dispersion was shaken at 150 rpm and incubated at 37 °C. Particles in the dispersion were removed using centrifugal separation and ultrafiltration. Specifically, centrifugation was conducted at 6840× *g* (i.e., 7080 rpm) for 15 min to precipitate particles larger than approximately 450 nm. A portion of the supernatant was further centrifuged for 60 min using a centrifugal ultrafiltration filter unit (Amicon Ultra; Sigma Aldrich), and the filtrate was used as the sample. A 3 kDa ultrafiltration filter (equivalent to a diameter of 1.5–3.0 nm) was used, and the collected filtrate was diluted 200× with ultrapure water containing a final concentration of 0.5 M nitric acid (HNO_3_). Afterward, the nickel was quantified using an inductively coupled plasma mass spectrometer (Agilent 8800; Agilent Technologies, Santa Clara, CA, USA). The eluted nickel content was calculated from the weight of each particle, which was determined when preparing the dispersion.

### 2.3. Experimental Animals

All animal experiments were performed on 6–8-week-old female BALB/c mice (CLEA, Tokyo, Japan). All mice were preliminarily bred for 1 week and fed normal feed and tap water ad libitum with a lighting cycle of 12 h per day.

### 2.4. Pharmacokinetic Evaluation

#### 2.4.1. Twenty-Eight-Day Repeated Oral Administration Test

We conducted a 28-day repeated oral administration test of the test substance to evaluate the pharmacokinetics of the Ni-NPs. The mice were divided among the following groups: Ni-NP dispersion administration group (Ni-NPs group); nickel microparticle dispersion administration group (Ni-MPs group); nickel chloride (NiCl_2_) administration group, where nickel ions were assumed (Ion group); and PBS administration group (PBS group). Tests were conducted on these groups (*n* = 5). We orally administered 100 µL of each particle dispersion, nickel chloride solution, or PBS once per day for 28 days. The nickel content per dose was 50 mg/kg (body weight) in the Ni-NPs and Ni-MPs groups and 5 mg/kg (body weight) for NiCl_2_. All mice were euthanized 24 h after the final dose, and whole blood and tissues were retrieved immediately.

#### 2.4.2. Histopathological Evaluation

The mice were anesthetized via isoflurane inhalation anesthesia 24 h after the final dose and then euthanized under comfortable conditions using CO_2_ gas. Thoracotomy and laparotomy were conducted immediately after euthanizing the animals, and a portion of the small intestine 10 mm from the origin of the duodenum in the direction of the anus was excised from each mouse. To prevent poor fixation, the excised small intestine was immersed in PBS and fixed with 10% formalin in 0.1 M phosphate buffer (pH 7.4) for 12 h without damaging the tissue. The tissue was then washed with PBS (10 min × 3 times), dehydrated in an ascending ethanol series, and embedded in paraffin according to the standard method. The embedded paraffin blocks were sliced to 2–3 µm, de-paraffinized with xylene, rehydrated, and stained with hematoxylin and eosin. After dehydration and clearing with xylene, the cells were sealed and observed with an optical microscope.

#### 2.4.3. Quantitative Evaluation of Nickel Accumulation

Part of the liver and the entire kidney were removed to quantitatively evaluate the amount of nickel that accumulated in each organ. The wet weight of each excised tissue was immediately measured using a precision balance to avoid weight changes due to drying. Organ samples were lysed with an acid lysis system (Digiprep Jr.; SCP Science, Baie-d’Urfé, QC, Canada). Specifically, each organ for which the wet weight was measured was placed in a dedicated flat-bottomed 50 mL tube (Digi Tube; SCP Science), after which 5 mL of 60% concentrated nitric acid was added, a watch glass was placed on the sample, and heat treatment was conducted. Heat treatment was conducted at 80 °C for 4 h and then at 105 °C to evaporate the nitric acid to the point of exhaustion. Afterward, 0.5 M of nitric acid was added, and organic matter was completely removed through filtering with a 0.45 µm filter unit (Millex; Sigma Aldrich); this was used as a sample. The dissolved samples were subjected to inductively coupled plasma mass spectrometry, and nickel analysis was conducted using the calibration curve method.

#### 2.4.4. Transmission Electron Microscopy

A part of the mouse liver tissue was used as a sample for transmission electron microscopy. After excising the tissue, it was immersed in 2% glutaraldehyde/0.1 M phosphate buffer (pH 7.4), cut into squares measuring approximately 1 mm2, and fixed on ice for 240 min (pre-fixation). After pre-fixation, the tissue was immersed in 0.1 M phosphate buffer for 60 min, washed, and similarly immersed in 1% osmium tetroxide solution and 0.1 M phosphate buffer (pH 7.4) for 120 min, until the tissue was completely blackened (post-fixation). The post-fixed blocks were dehydrated in an ascending ethanol series, which was subsequently replaced with 100% propylene oxide, and then immersed in a mixed solution of epoxy resin (EPON812; TAAB Laboratories Equipment, Berkshire, UK) and propylene oxide, with the concentration of epoxy resin increased in a sequential manner. Finally, the block was immersed in 100% epoxy resin and allowed to stand at 35 °C for 12 h to allow the resin to permeate the block. The block was then left to stand at 45 °C for 24 h to dry the substitute agent remaining in the resin and was embedded after heat polymerization treatment at 60 °C for 2 days. For the embedded resin block, semi-ultrathin sections were prepared using an ultramicrotome (ULTRACUT; Leica, Wetzlar, Germany) and stained with toluidine blue. After confirming the state of the tissue, ultra-thin sections of 70 nm were prepared, subjected to electron staining with uranyl acetate and lead citrate, and observed with a transmission electron microscope (H-7100; Hitachi, Tokyo, Japan).

#### 2.4.5. Measurement of Serum Cytokine Concentrations

Whole blood was collected via cardiac puncture after the mice were euthanized. Blood was sampled using a 1 mL tuberculin syringe with a 26-gauge needle and retrieved in a micro-blood container (Minicorrect II; Greiner Bio-one, Monroe, NC, USA). The blood collected from the mice was centrifuged at 3000 rpm and 4 °C for 10 min, after which the serum was sampled. The serum was stored at −20 °C, thawed at 4 °C immediately before performing the ELISA, and then gradually returned to room temperature. Quantitative evaluation of the interleukin (IL)-17 and IL-1β levels in the serum from all mice was conducted using the Mouse IL-1β, IL-17 ELISA kit (R&D Systems, Minneapolis, MN, USA). All procedures were conducted according to the manufacturer’s manual. After conducting the ELISA, the absorbance of all wells (primary wavelength of 450 nm and reference wavelength of 620 nm) was measured using an absorptiometer (Infinite^®^ F200; TECAN, Männedorf, Switzerland) and converted to pg/mL.

### 2.5. Evaluation of Allergenicity

#### 2.5.1. Production of Nickel Allergy-Model Mice

In generating nickel allergy-model mice, 10.0 µg/mL lipopolysaccharide from Escherichia coli serotype 055: B5 (LPS; Sigma Aldrich) was used as an adjuvant for sensitization. We intraperitoneally administered 250 µL of the Ni-NP or Ni-MP dispersion or 5.0 mM NiCl_2_ solution to induce sensitization to nickel. Intraperitoneal administration was conducted using a 26 G needle. Only 10.0 µg/mL of LPS solution was administered to the PBS group. Seven days after sensitization, 20 µL of a 5 mM nickel chloride solution was administered intradermally to the right auricle with animals under isoflurane inhalation anesthesia to elicit nickel allergy. A 33 G needle was used for intradermal administration. Considering that nickel allergy is a systemic immune reaction, the drug was administered intradermally to only one side and not to both sides (n = 5).

#### 2.5.2. Measurement of Auricle Thickness

After nickel allergy elicitation, the auricle thickness was measured at regular intervals. Measurements were conducted with animals under isoflurane inhalation anesthesia using a dial thickness gauge (Peacock; OZAKI, Tokyo, Japan), and the average values of the three measurements at each time point were used as the data. Auricular swelling was calculated as the increase in ear thickness after elicitation relative to the thickness immediately before elicitation (mean ± standard deviation). After measuring the auricle thickness 72 h later, the mice were euthanized, whole blood was collected, and the right auricle was excised.

#### 2.5.3. Histopathological Evaluation

The excised mouse auricles were embedded in paraffin blocks as described in Section 2.4.2, sectioned, stained with hematoxylin and eosin, and examined microscopically.

#### 2.5.4. Measurement of Serum Cytokine Concentration

IL-17 and IL-6 in the collected serum were measured using ELISA (R&D Systems, Minneapolis, MN, USA). Afterward, the concentration was converted as described in Section 2.4.5.

#### 2.5.5. Statistical Analysis

Statistical analysis was conducted using statistical analysis software (SPSS Statics 25; Chicago, IL, USA). Student’s *t*-test or one-way analysis of variance was used to test the difference in the means of each group, and Tukey’s test was used for subsequent multiple comparisons. Differences were considered significant at *p* < 0.05.

## 3. Results

### 3.1. Evaluation of Particle Characteristics

#### 3.1.1. Morphological Observation and Elemental Analysis

A film-like structure was confirmed to have formed on the surface of Ni-NPs as noted via transmission electron microscopy. We then conducted image-based particle size measurements on 100 particles each comprising Ni-NPs and Ni-MPs, which were randomly extracted, and results showed that the average primary particle sizes were 22.3 ± 8.0 nm for Ni-NPs and 2796.0 ± 1471.7 nm for Ni-MPs (Figure 1a,b). Results of scanning electron microscopy showed that the Ni-NPs had a spherical and smooth surface, whereas the Ni-MPs had a spherical or cubic crystal structure with a rough surface (Figure 1c,d). EDX analysis showed a nickel peak for all particles, but Ni-NPs exhibited an enhanced oxygen peak as well (not shown in figure).

#### 3.1.2. Calculation of Specific Surface Area

Nitrogen was adsorbed onto the surface of each particle using a gas adsorption device, and the specific surface area (S_BET_) was calculated from the adsorption amount. Results showed that the values were 13.89 m^2^/g for Ni-NPs and 1.34 m^2^/g for Ni-MPs, with the former having a specific surface area approximately 10.4-fold greater than that of Ni-MPs (Table 1).

#### 3.1.3. Particle Size Distribution Measurement

Particle size analysis (secondary particle size measurement) was conducted using a particle size distribution analyzer immediately after dispersing each particle in ultrapure water using an ultrasonic homogenizer. Results showed that each particle had an aggregated morphology, and the mode of particle sizes was 170.0 nm for Ni-NPs and 11,890.0 nm for Ni-MPs, indicating 7.8-fold and 4.3-fold greater aggregation compared to the primary particle size, respectively.

#### 3.1.4. Ion Elution Test

Inductively coupled plasma mass spectrometry was used to measure the nickel ion elution content over time for each particle dispersion. Results showed that more nickel ions were eluted from Ni-MPs than from Ni-NPs at 12 and 24 h after preparation of the dispersion. After 24 h, 5.00 ± 0.16% of Ni-MPs and 1.92 ± 0.04% of Ni-NPs were ionized from the total amount of dispersed particles (Figure 2).

### 3.2. Evaluation of Pharmacokinetics

#### 3.2.1. Twenty-Eight-Day Repeated Oral Dose Study

We used 20 BALB/c female mice for a 28-day repeated oral administration study of each test substance. No mice died during the experimental period. There were also no statistical differences in body weight change between the groups during the experimental period.

#### 3.2.2. Histopathological Evaluation

Figure 3 shows the histological images of the mouse small intestine stained with hematoxylin and eosin after a 28-day repeated oral administration test. Compared to the PBS group, the Ion group showed significant atrophy and degeneration of the glandular structure at the base of the epithelium, and the Ni-NPs group also lacked continuity in the glandular structure. Meanwhile, no significant histological findings were obtained with the Ni-MPs group.

#### 3.2.3. Quantitative Evaluation of Nickel Accumulation

Figure 4 shows the results of the quantitative evaluation of the nickel accumulation in the tissue, which was analyzed using an inductively coupled plasma mass spectrometer. In all groups that were administered nickel, the nickel amount in the liver and kidney increased, but the accumulation was significantly higher in the Ion and Ni-NPs groups than in the PBS group. Meanwhile, there were no differences observed between the Ni-MPs and PBS groups.

#### 3.2.4. Transmission Electron Microscopy

Figure 5 shows the results of observations, via transmission electron microscopy, of the liver tissue in the Ni-NPs group. Results of a detailed observation of the pharmacokinetics of Ni-NPs were as follows. Based on low-magnification images of the Ni-NPs group (Figure 5a), an accumulation of electron-dense particle aggregates was observed inside the phagolysosomes of sinusoidal endothelial cells, which constitute countless cavities (sinusoids), in liver tissue. Based on high-magnification images (Figure 5b), there was an uptake image of Ni-NPs with a size and shape similar to those of the administered Ni-NPs. Additionally, for the Ion group, based on low-magnification images (Figure 5c), there was also an accumulation of electron-dense particle aggregates in the Disse space, which is an intermediate segment between hepatocytes and sinusoids; moreover, in the high-magnification images (Figure 5d), aggregates of Ni-NPs with a cubic crystal structure were observed.

#### 3.2.5. Measurement of Serum Cytokine Concentrations

Figure 6 shows the concentrations of each cytokine in the serum, quantified using the ELISA method. The serum IL-17 concentrations were significantly higher in the Ion (20.0 ± 5.5 pg) and Ni-NPs (16.5 ± 8.1 pg) groups than in the PBS group (below detection limit) and in the Ion and Ni-NPs groups than in the Ni-MPs group (1.0 ± 4.5 pg) (Figure 6a). Additionally, the serum IL-1β concentrations were significantly higher in the Ion (20.7 ± 6.7 pg) and Ni-NPs (18.0 ± 7.4 pg) groups than in the PBS group (7.4 ± 1.6 pg) (Figure 6b). No statistical differences were observed between the Ni-MPs (11.3 ± 2.9 pg) and PBS groups based on all results.

### 3.3. Evaluation of Allergenicity

#### 3.3.1. Measurement of Auricle Thickness

We used 20 BALB/c female mice to evaluate allergenicity, as in the 28-day repeated oral administration test. No mice died during the experimental period. Figure 7 shows the results of measurements of the auricle thickness in mice that underwent local elicitation of nickel allergy in the right auricle every 24 h and a comparison of swelling levels. Based on changes in auricle thickness in the PBS group, it was determined that the effects of LPS and fluid volume on the thickness were negligible. In the three groups (Ion, Ni-NPs, and Ni-MPs groups), which were sensitized to nickel, ear thickening increased after elicitation, peaked at 48 h, and then decreased. At 48 h after elicitation, the Ni-MPs group (44.0 ± 2.8 µm) showed a significantly higher value than the PBS group (3.3 ± 5.3 µm), and the Ion (83.3 ± 2.4 µm) and Ni-NPs (84.7 ± 2.4 µm) groups showed significantly higher values than the Ni-MPs group.

#### 3.3.2. Histopathological Evaluation

Figure 8 shows the results of hematoxylin and eosin staining and histopathological observations of the right auricle of each group of mice 72 h after nickel allergy elicitation. The histology was consistent with the presence of auricular cartilage in the center and the inner skin (insertion side) above the auricular cartilage. Increased infiltration of chronic inflammatory cells, mainly lymphocytes, was observed in the Ion, Ni-NPs, and Ni-MPs groups compared to that in the PBS group, but this was particularly pronounced in the Ni-NPs group. Lymphocyte infiltration in the Ni-MPs group was minimal.

#### 3.3.3. Measurement of Serum Cytokine Concentrations

Figure 9 shows the concentrations of each cytokine in the serum as quantified using the ELISA. The serum IL-17 concentrations were significantly higher in the Ion (4.5 ± 2.5 pg) and Ni-NPs (6.0 ± 3.0 pg) groups than in the PBS group (below detection limit) (Figure 9a). Similarly, the serum IL-6 concentrations were significantly higher in the Ion (5.2 ± 4.3 pg) and Ni-NPs (9.3 ± 2.3 pg) groups than in the PBS group (below detection limit) (Figure 9b). No statistically significant differences were observed between the Ni-MPs and PBS groups based on any of the results.

## 4. Discussion

### 4.1. Particle Characteristics

In this study, we focused on nickel, which is frequently used in dental treatment and epidemiologically is associated with a high incidence of metal allergy; specifically, we investigated the pharmacokinetics associated with oral exposure and its involvement in the development of allergies to elucidate the effects of Ni-NPs on the living body. When investigating the effects of nanoparticles on living organisms, it is essential to evaluate the physical properties of the particles being used. It was confirmed via the EDX analysis of Ni-NPs and Ni-MPs used in this study that Ni-NPs had more emphasized oxygen peaks than Ni-MPs, and thus, it was thought that the film structure on the Ni-NP surface was an oxide film. In the comparison of each particle based on transmission electron microscopy images, although the magnification was different, the metal surface usually had an oxide film, and thus, it was thought that the oxide film was also present on the surfaces of the Ni-MPs, but the difference in EDX peak intensity suggested that the oxide layer in Ni-NPs was thicker than that in Ni-MPs. However, these findings were mainly derived from the characterization of powder samples, and it is possible that the characteristics differ from those of the dispersion liquids that were administered. Therefore, particle size distribution measurements were conducted to evaluate PBS dispersibility, and the nickel ion content released over time was measured via inductively coupled plasma mass spectrometry (Figure 2). Based on a comparison of the average particle size calculated via transmission electron microscopy, the mode of the obtained particle size distribution was approximately 7.80-fold greater than the particle size of the Ni-NPs and approximately 4.3-fold greater than the particle size of the Ni-MPs, suggesting the formation of aggregates. Previous studies have reported that nanoparticles usually form aggregates approximately 10 times larger than the primary particle size even when subjected to ultrasonic treatment in a solution, consistent with the results of the present study [18,19]. The prepared dispersion was air-dried immediately, and the dispersibility was confirmed based on the image using a scanning electron microscope, consistent with the results obtained through the particle distribution measurements. Additionally, measurements of the nickel ion elution content in each particle dispersion require the removal all fine particles in the dispersion. Normally, nanoparticles with a diameter of 10 nm or less are difficult to separate via centrifugation because Brownian motion affects particle falling [20]. Therefore, an ultrafiltration filter was used to remove particles. Ultrafiltration filters are defined by their molecular weight, which is not consistent with the nanoparticle size. Therefore, nanoparticles were removed according to a protocol previously provided in the literature [21]. The ions eluted from particles were generally thought to exhibit improved surface reactivity, with the release of more ions as the particle size decreases and the specific surface area increases [22]. However, when comparing the nickel ion elution content from each particle in this study, it was shown that the value for the Ni-MPs exceeded that for the Ni-NPs. This was thought to be due to the difference in the thickness of the surface oxide film layer, but the maximum ionization rate of all particles after 24 h was approximately 5.0%, indicating that both particles were stable in the dispersion liquid. There was a concern that the difference in the thickness of the oxide film layer on each particle might affect subsequent experiments; however, a previous study has reported that the presence of the oxide film layer does not affect the toxicity of Ni-NPs [23].

### 4.2. Evaluation of Pharmacokinetics

Toxicity comparisons and pharmacokinetics studies of Ni-NPs and Ni-MPs were conducted as described in previous reports [24,25]. Rats are usually the animal species of choice in such experimental systems, and although equal numbers of male and female animals are used in these experiments, we used only female mice in the present study. The reason for this is related to the epidemiology of nickel allergy, discussed in the introduction. In humans, the prevalence of nickel allergy is higher in females than in males, and previous studies have often used nickel allergy-model mice as the animal species [26]. Epidemiological research has indicated that the sex-based difference in nickel allergy prevalence is simply due to women being exposed to more metals daily than men, such as with necklaces and jewelry [27]. In the present study, only female mice were used to investigate the effects of Ni-NPs on the immune system through a comparison of repeated oral administration test models and nickel allergy-model mice.

Numerous reports have been published on genotoxicity and changes in tissues of various organs based on repeated oral administration tests of Ni-NPs, but few reports have compared these with fine particles and even fewer reports have investigated their effects on the intestinal tract [28,29]. The results of histological observations of the small intestine stained with hematoxylin and eosin in this study showed that the glandular structures in the Ion group were arranged in the epithelial basal part, including Paneth cells, and showed significant atrophy and degeneration, and this part was also partially missing in the Ni-NPs group (Figure 3b,c). Meanwhile, there were no notable findings in the Ni-MPs group, as compared with observations of the PBS group (normal tissue) (Figure 3a,d). Based on the results of nickel measurements in the liver and kidney via inductively coupled plasma mass spectrometry, the Ion and Ni-NPs groups had significantly higher values than the PBS group, but there was no difference between the PBS and Ni-MPs groups (Figure 4a,b). Furthermore, serum IL-1β and IL-17 concentrations were significantly higher in the Ion and Ni-NPs groups than in the PBS group, and in particular, IL-17 concentration was not significantly increased in the Ni-MPs group (Figure 6a,b). In the repeated oral administration test, considering the toxicity of the nickel ions, the nickel content used in the Ion group was set to approximately 1/10 that in the Ni-NPs and Ni-MPs groups, but strong inflammatory reactions nevertheless occurred. These results suggest that the toxicity of nickel is mainly caused by ions, which is consistent with the results reported in previous studies [30,31]. When comparing the Ni-NPs and Ni-MPs groups, inflammation was clearly more pronounced in the Ni-NPs group. Assuming that the nickel toxicity is caused only by ions, in the characteristic evaluation, the Ni-MPs—which had more nickel ions eluted—should produce a strong inflammatory reaction, but this was not consistent with the present results. This suggests that the main cause of toxicity in the Ni-NPs group was not the nickel ions but rather the Ni-NPs.

The accumulation of Ni-NPs in the liver and kidney is thought to occur via cells such as macrophages that have phagocytosed the particles or via endocytosis [32]. Among them, regarding the uptake of nanoparticles via endocytosis, it has been reported that particles with a size of 1 µm or greater are not taken up by cells, whereas nano-sized particles are taken up into cells via receptors [33]. All of these results are based on in vitro culture, and the possibility that the penetration rate of particles would increase in vivo due to damage to the intestinal epithelium needs to be considered. Although the results of this study do not provide sufficient evidence to support these findings, previous reports have suggested that Ni-NPs affect epithelial tight junction-associated proteins [34].

To investigate the detailed dynamics of Ni-NP accumulation in the body after repeated oral administration, we performed transmission electron microscopy of the liver sections. The reason for focusing on the liver is that it is the major organ where metal nanoparticles accumulate and exert toxic effects [30]. Double staining with uranyl acetate and lead citrate is usually used in transmission electron microscopy to enhance contrast. Electron staining clarifies the membrane structures of organelles, which are usually difficult to observe, but it is possible that this might make it difficult to distinguish metal nanoparticles. Therefore, we first compared unstained sections (which show some ability to be stained owing to the osmium tetroxide used as a post-fixative) and electron-stained sections to confirm the visibility of the accumulated Ni-NPs. Based on this comparison, it was confirmed that the electron beam impermeability of the Ni-NPs was enhanced due to the high electron density, and accumulation in the hepatocyte nuclei was observed. Normal observations were made after confirming this. In the Ni-NPs group, the accumulation of Ni-NPs in the liver within phagolysosomes of sinusoidal endothelial cells lining the lumen of sinusoids, known as intrahepatic capillaries, was observed (Figure 5a,b). In the process by which cells take up substances inside and outside the cell (i.e., endocytosis), phagosomes exhibit so-called phagocytosis, and phagolysosomes combine with lysosomes containing digestive enzymes to digest the ingested substances. Previous reports have shown that metal nanoparticles exhibit high solubility in artificial lysosomal fluid [35]. Ni-NPs taken up by cells are lysed via mechanisms such as those associated with phagolysosomes, and an increase in the intracellular nickel concentration is thought to result in cytotoxicity. Endocytosis also occurs in the immune system, which functions in the presentation of a portion of the degraded substance as the antigen [36]. Metal nanoparticles are taken up into cells through these pathways, and inflammatory signals are activated to induce cell membrane damage and apoptosis. Furthermore, the incorporated nanoparticles are secreted from the cell via exocytosis, which is an extracellular secretion mechanism, and they spread in the vicinity of the cell. This sequence of events has been suggested by researchers to cause toxicity through a Trojan horse-like mechanism [37,38]. The incorporation of Ni-NPs into phagolysosomes indicates that nickel might have been presented as an antigen via either route. Importantly, the accumulation of Ni-NPs with a crystalline structure was also confirmed in liver tissue of the Ion group (Figure 5c,d). This indicates the possibility that nickel, which was administered as ions, formed nanoparticles in the body. There are few reports on orally administered metal ions taking the form of nanoparticles in major organs of the body, but it has been reported that silver nanoparticles that are administered into the blood tend to be distributed in the form of particles in the liver and in the form of ions in the blood [39]. In the study by Ishizaka T et al. [39], the research model comprised single-particle inductively coupled plasma mass spectrometry to collect nanoparticles from living tissue and perform qualitative and quantitative analysis, and it was reported that nanoparticles could be clearly detected in tissues of the group that received silver ions. However, this comprised an indirect detection method, and in the present study, we were able to show the formation of nanoparticles from ions based on our imaging findings. The mechanism of Ni-NP formation in the body is unknown, but nickel shows poor solubility in pure water. However, in the presence of chloride ions, it exhibits corrosive behavior, and its dissolution rate increases [40]. In vivo, chloride ions are abundant in extracellular fluids, such as plasma and interstitial fluid, whereas they are scarcely present in intracellular fluids [41]. This is probably because nickel is abundantly present as ions in the blood and in the intercellular space, and nickel taken into the cells accumulates as particles. In other words, these results suggest that nickel is constantly changing from ions to nanoparticles, as well as from nanoparticles to ions, in vivo. However, there are still many unknowns with respect to this process, and further research is needed to clarify the pharmacokinetics of metal nanoparticles.

### 4.3. Evaluation of Allergenicity

The protocol for creating the nickel allergy model was based on previous reports, and the final concentration was determined in a preliminary experiment [42]. Many substances are used as adjuvants to induce a nickel allergy in mice, but in the present study, *E. coli*-derived LPS was used. This is because, according to previous reports, it has minimal effects on the body compared to that by other adjuvants [43]. Furthermore, LPS is also known as a cell wall component of gram-negative bacteria, which are oral bacteria, and it is an important factor in the experimental model of the present study, in which the effects of oral exposure to nickel are assumed. Many metals such as nickel, cobalt, and chromium have been reported to induce allergic reactions in mice when combined with LPS, and in humans, it might promote the development of allergies and enhance inflammatory reactions [44].

In the present study, the increase in auricle thickening in the Ion, Ni-NPs, and Ni-MPs groups indicated that sensitization to nickel induced a nickel allergic reaction (Figure 7). An observation of the auricular histology 72 h after nickel allergy elicitation revealed chronic inflammatory cell infiltration mainly composed of lymphocytes in the Ion, Ni-NPs, and Ni-MPs groups. Inflammatory cell infiltration was particularly prominent in the Ion group (Figure 8). Furthermore, serum cytokine analysis revealed that IL-6, which is a representative inflammatory cytokine, was increased significantly in the Ion and Ni-NPs groups (Figure 9). Given the results of the characteristic test, we compared the Ni-NPs and Ni-MPs groups despite the fact that the nickel elution content in PBS was large in the Ni-NPs group, and the results showed that the inflammatory response in the Ni-NPs group tended to be exacerbated and the auricle thickness was similar to that in the Ion group. This indicates that Ni-NPs have the same sensitization ability as nickel ions. The detailed mechanism underlying metal allergy is still unknown, but it is thought that nickel as an ion does not result in direct sensitization and that it becomes a hapten by binding to tissue proteins such as albumin to exhibit immunogenicity [45]. However, in recent years, it has been shown that nickel directly binds to the MHC complex that is present on the surface of lymphocytes and activates T cells [46]. The details of the effects of Ni-NPs on lymphocytes are currently unknown, but it is possible that they might exert direct effects similar to those of nickel ions, and further research is needed to address this.

In the repeated oral administration and nickel allergy elicitation studies, both Ion and Ni-NPs groups showed significant increases in IL-17. IL-17 comprises a novel cytokine family, discovered in 1995, which acts on cells such as epithelial cells and macrophages, and inflammation is induced through the induction of inflammatory cytokines such as IL-6 and tumor necrosis factor-α. IL-17-producing cells include Th17 cells, a type of CD4-positive T cell, but it is also known to be produced by various other cells such as CD8-positive T cells and macrophages [47]. A recent report indicated that TH17 cells play an important role in the development of type IV allergies, including metal allergy [17]. The increase in serum IL-17 concentrations in the nickel allergy elicitation test, using nickel allergy-model mice of the Ion and Ni-NPs groups, showed that the inflammatory reaction induced by nickel was due to IL-17. Moreover, the increase in IL-17 concentrations in the repeated oral administration test suggested the possibility that Th17 cells were activated and established sensitization to nickel. However, in the nickel allergy elicitation study, nickel and LPS were administered intraperitoneally at the time of sensitization, and it should be noted that an increase in the IL-17 concentration is not due to auricle inflammation alone.

The mechanism by which nickel nanoparticles contribute to the development of allergies is discussed. It has been suggested that nickel nanoparticles accumulate in tissues for longer periods of time than regular particles due to their higher frequency of uptake into cells and intracellular organelles [35]. In contrast to larger particles, nanoparticles can easily pass through cell membranes via receptors, leading to an increase in intracellular nickel concentration and induction of apoptosis [33]. The released nanoparticles have been found to diffuse extracellularly via exosomes, promoting signaling to the immune system and activating dendritic cells and T cells [38]. This study further supports these reports and reinforces their findings. However, further research is needed to fully elucidate the detailed mechanism of Ni-NP-induced allergic shedding.

### 4.4. Summary of Results

The results of the present study indicate that oral exposure to Ni-NPs, as opposed to Ni-MPs, causes intestinal disturbances and that these particles accumulate in large quantities in the major organs (e.g., the liver and kidney). The fact that Ni-NPs were also observed in the nickel ion-administration group indicates that nickel might constantly change from ions to nanoparticles and from nanoparticles to ions in vivo, and the incorporation of Ni-NPs into the phagocytic cells suggests an effect on the immune system via exosomes. The re-administration of nickel to mice that were treated with Ni-NPs elicited an allergic reaction, suggesting that Ni-NPs caused sensitization in the same manner as that with nickel ions, further suggesting that Th17 cells are involved in the onset of nickel allergy.

It should be noted that the route and amount of nickel exposure in this study do not directly reflect general daily life or dental practice. In our study, we administered relatively large doses of nickel to mice to investigate the bioavailability and allergenicity of nickel nanoparticles within a limited sample size and study period. The findings from this study are presented as basic data for considering the mechanism of health hazards posed by nickel nanoparticles.

In order to decrease the impact of potential risk for allergic reaction due to nickel-containing alloys, use of titanium alloys with greater biocompatibility is considered as the best option in clinical orthodontics.

## 5. Conclusions

Under the predetermined circumstances and situational constraints that were imposed for the purposes of this examination, oral exposure to Ni-NPs resulted in more serious adverse effects and bioaccumulation than that of Ni-MPs, suggesting the possibility of inducing allergies. Thus, we suggest that Th17 cells might be involved in the onset of nickel allergy; however, further research is needed to clarify the associated mechanism in depth.

## Figures and Tables

**Figure 1 materials-16-01834-f001:**
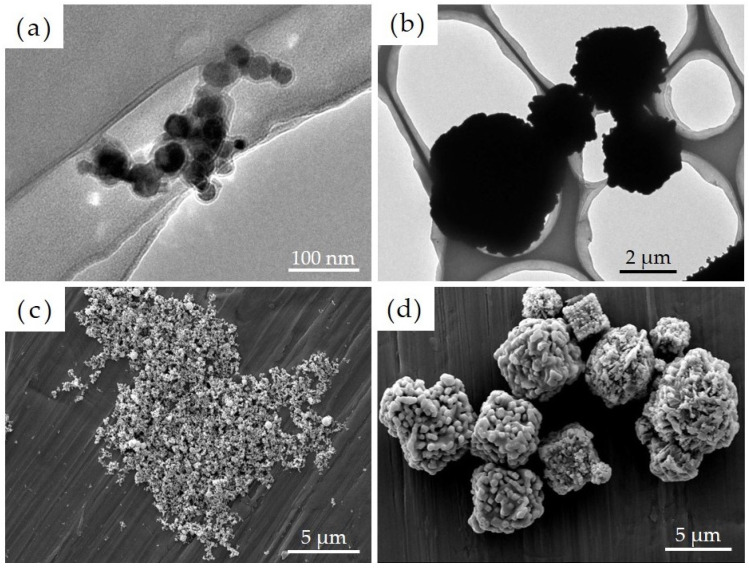
Transmission electron microscope images of each particle showing (**a**) nickel nanoparticles (Ni-NPs) and (**b**) nickel microparticles (Ni-MPs). Scanning electron microscope images of (**c**) Ni-NPs and (**d**) Ni-MPs are also shown.

**Figure 2 materials-16-01834-f002:**
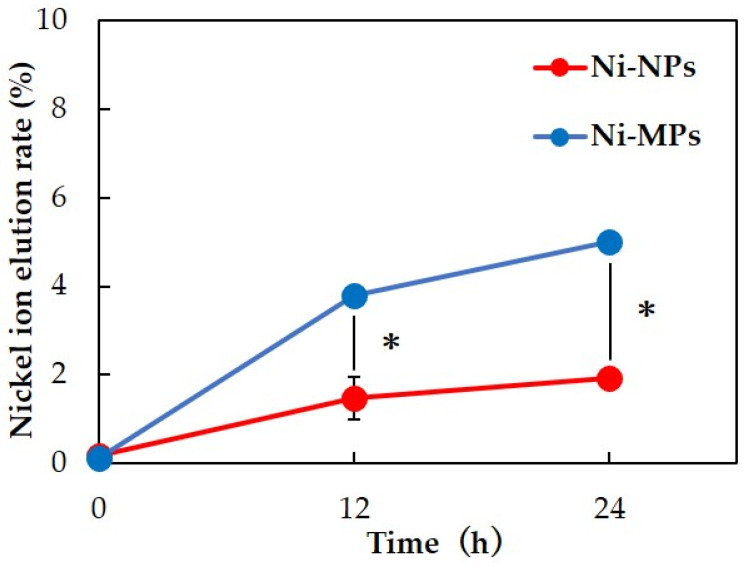
Comparison of nickel ion elution rate over time for each particle dispersion. Data are presented as the mean ± S.D., * denotes statistical significance *p* < 0.05.

**Figure 3 materials-16-01834-f003:**
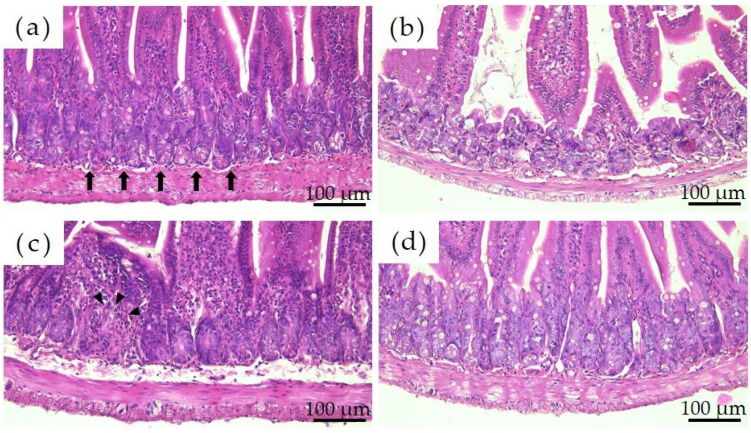
Comparison of the histology of the small intestine after repeated oral administration test. (**a**) PBS, (**b**) Ion, (**c**) nickel nanoparticles (Ni-NPs), and (**d**) nickel microparticles (Ni-MPs) groups. Histological observation based on hematoxylin and eosin staining showed significant atrophy and degeneration of glandular structures (black arrows) at the base of the epithelium in the Ion group compared with that observed in the PBS group. There was also some lack of continuity in the Ni-NPs group (black arrowheads). No notable changes were observed in the Ni-MPs group.

**Figure 4 materials-16-01834-f004:**
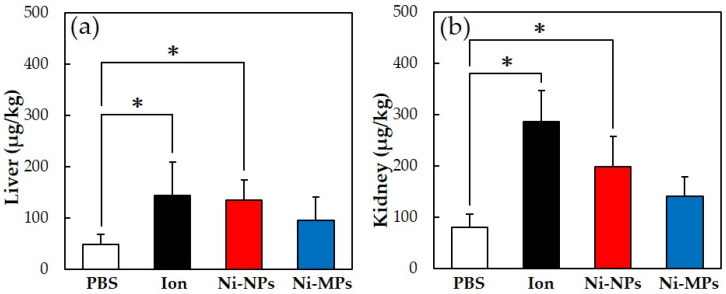
Comparison of nickel accumulation content in each organ after repeated oral administration test. (**a**) Liver, (**b**) kidney. Data are presented as the mean ± S.D., * denotes statistical significance *p* < 0.05.

**Figure 5 materials-16-01834-f005:**
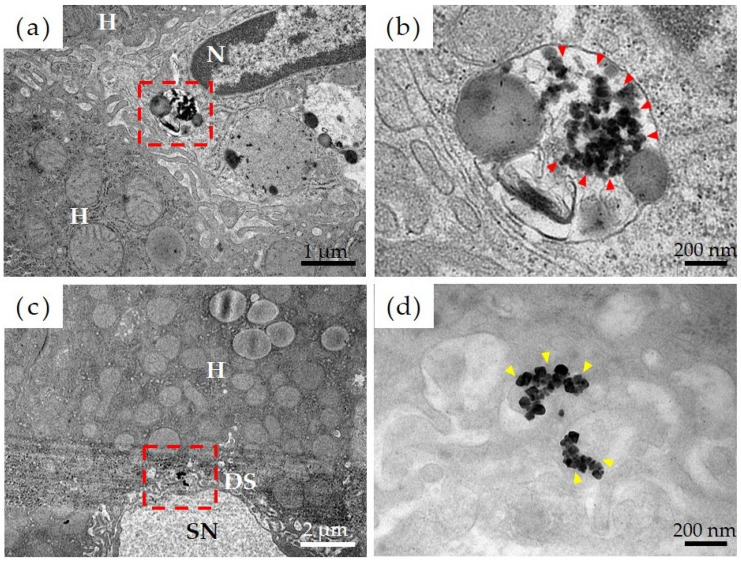
Evaluation of pharmacokinetics of nickel nanoparticles (Ni-NPs) via transmission electron microscopy. (**a**,**b**) Ni-NPs group, (**c**,**d**) Ion group; (**a**,**c**) low-power magnification image, and (**b**,**d**) high-power magnification image. The red frame in the low-power magnification image corresponds to the field of view from the high-power magnification image. The accumulation of Ni-NPs in phagolysosomes was observed in the Ni-NPs group (red arrowhead). Additionally, in the Ion group, the accumulation of Ni-NPs with a crystalline structure was observed (yellow arrowhead). H: hepatocytes, N: nuclei, SN: sinusoids; DS: Disse space.

**Figure 6 materials-16-01834-f006:**
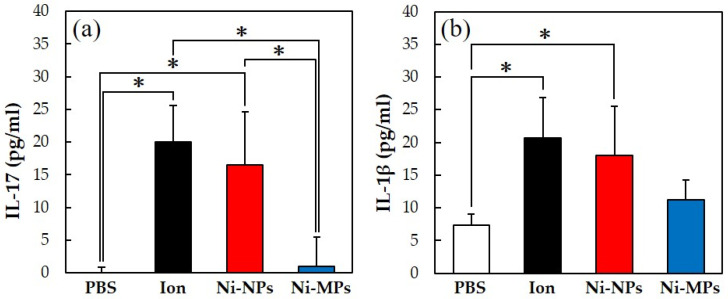
Comparison of serum cytokine levels after repeated oral administration test. (**a**) IL-17, (**b**) IL-1β. Data are presented as the mean ± S.D., * denotes statistical significance *p* < 0.05.

**Figure 7 materials-16-01834-f007:**
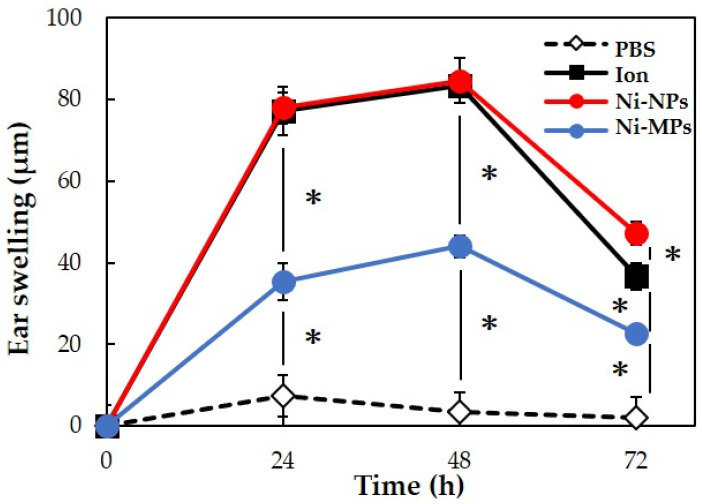
Comparison of auricular swelling after nickel allergy elicitation. Data are presented as the mean ± S.D. * denotes statistical significance *p* < 0.05.

**Figure 8 materials-16-01834-f008:**
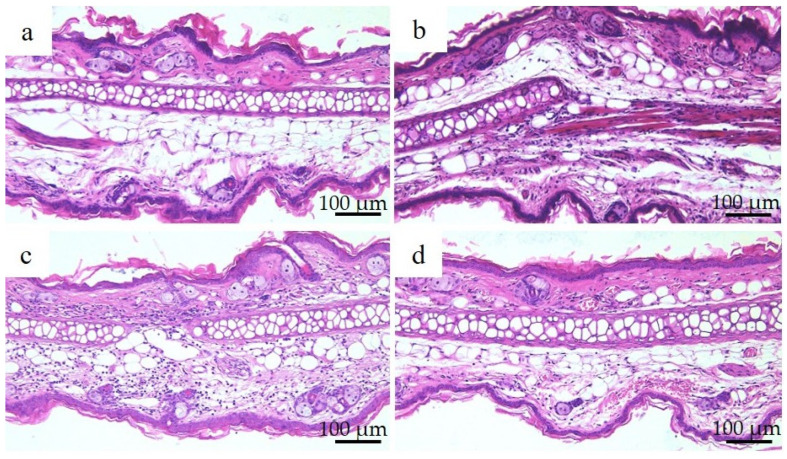
Auricular histology in each group after nickel allergy elicitation. (**a**) PBS, (**b**) Ion, (**c**) nickel nanoparticle (Ni-NPs), and (**d**) nickel microparticle (Ni-MPs) groups. Results of histological observations based on hematoxylin and eosin staining showed chronic inflammatory cell infiltration, which was mainly composed of lymphocytes in the Ion, Ni-NPs, and Ni-MPs groups, which was particularly pronounced in the Ni-NPs group.

**Figure 9 materials-16-01834-f009:**
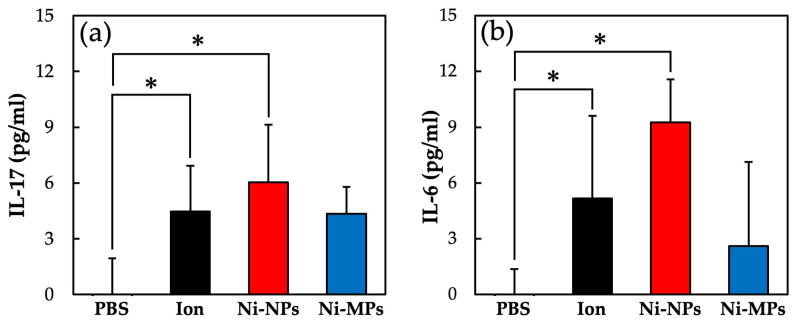
Comparison of serum cytokine levels after nickel allergy elicitation. (**a**) IL-17 and (**b**) IL-6. Data are presented as the mean ± S.D., * denotes statistical significance *p* < 0.05.

**Table 1 materials-16-01834-t001:** Summary of the characteristics of each particle.

	Ni-NPs	Ni-MPs
Shape Surface properties	Spherical Smooth Oxide film structure	Spherical or cubic Rough
Primary particle size	22.3 ± 8.0 nm	2796.0 ± 1471.7 nm
Secondary particle size	174.0 nm	11,890.0 nm
Aggregation rate (Secondary particle size/ primary particle size ratio)	7.8	4.3
Nickel ion elution rate (At 24 h)	1.92 ± 0.04%	5.00 ± 0.16%
Specific surface area S_BET_	13.89 m^2^/g	1.34 m^2^/g

Mean ± S.D.

## Data Availability

Not applicable.

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
