# Peer review of "Allergenicity and Bioavailability of Nickel Nanoparticles Compared to Nickel Microparticles in Mice"

_materials, 2023, doi:10.3390/ma16051834_

Round 1

Reviewer 1 Report

Thank you for submitting your work for publication in Materials Journal. This paper compares the Allergenicity and bioavailability of nickel nanoparticles to microparticles in mice model. The experimental deign is sound and the findings are presented well. 

There are some concerns in regards to the model itself. Its clinical implications are difficult to interpret mainly due to the manner in which the exposure of mice to the allergen was conducted. The oral or intraperitoneal route of administration cannot be compared to the dental use of such metals wherein release/leaching of ions into the circulation is very minimal. It would be highly unlikely to be similar in dosage or frequency when dental materials containing nickel are used.

The authors should discuss this aspect in detail within the discussion section. Additionally, proposing alternative metals/materials for dental use may be beneficial.

Other minor comments are available on the annotated PDF attached.

Author Response

Comments and Suggestions for Authors and Our Responses

Thank you for submitting your work for publication in Materials Journal. This paper compares the Allergenicity and bioavailability of nickel nanoparticles to microparticles in mice model. The experimental deign is sound and the findings are presented well. 

There are some concerns in regards to the model itself. Its clinical implications are difficult to interpret mainly due to the manner in which the exposure of mice to the allergen was conducted. The oral or intraperitoneal route of administration cannot be compared to the dental use of such metals wherein release/leaching of ions into the circulation is very minimal. It would be highly unlikely to be similar in dosage or frequency when dental materials containing nickel are used.

Our Response

We understand that our data from animal study does not support real human behavior. The objective of this research is to elucidate the in vivo dynamics of nanoparticles, which are extremely small structures, and their immune response. This study was designed in order to obtain concrete data proving the harmfulness, especially of Ni nanoparticles versus Ni microparticles. For reasonable interpretation and understanding by the readers, some of consideration and limitation of this study have been discussed in the discussion section. In addition, conclusion section has been rewritten (Page 16, Lines 590-598; Page 17, Lines 601-602).

The authors should discuss this aspect in detail within the discussion section. Additionally, proposing alternative metals/materials for dental use may be beneficial.

Our Response

In order to decreasing the impact of potential risk for allergic reaction due to nickel-containing alloys, use of titanium alloys with greater biocompatibility is considered as the best way in clinical orthodontics. The new sentences have been added in discussion section (Page 16, Lines 590-598; Page 16, Lines 596-598).

Other minor comments are available on the annotated PDF attached.

Our Response

All suggested points of the PDF have been corrected.

Thank you for giving us the opportunity to strengthen our manuscript with your valuable comments and queries. We have worked hard to incorporate your feedback and hope that these revisions persuade you to accept our submission.

Reviewer 2 Report

- Latin names of organisms are written in italics in international terminology.

- The ethicality of the research is questionable, considering the sample used. It may bother some.

- I would welcome a detailed chemical background of the allergic reaction mechanism

- Considering the content of the study, I would consider submitting it to another journal.

Author Response

Comments and Suggestions from Authors and Our Responses

- Latin names of organisms are written in italics in international terminology.

Our response

As suggested by reviewer, the text have been corrected in current version of our manuscript.

- The ethicality of the research is questionable, considering the sample used. It may bother some.

Our response

The objective of this research is to elucidate the in vivo dynamics of nanoparticles, which are extremely small structures, and their immune response. Investigating these phenomena in a single group of cells with only partial functions is not sufficient to prove our hypotheses, and animal studies are essential to investigate the complex immune response. Therefore, we submitted an application to our university's animal experimentation committee to seek approval for the ethics and appropriateness of such studies before conducting this research. Nickel is a metal with a high incidence of allergy, and given the rapid proliferation of nanotechnology, its safety is a growing concern. We believe that the findings obtained from this study will provide important basic data for the safe and effective future use of nanotechnology.

- I would welcome a detailed chemical background of the allergic reaction mechanism

Our response

To make this point clearer, we have added new paragraph in discussion section and allergic reaction mechanism has been discussed (Page 16, Lines 568-577).

- Considering the content of the study, I would consider submitting it to another journal.

Our response

Thank you for pointing out that our manuscript could be understood in this way. We believe that our research deal with biological reaction of nickel nanoparticle is in agreement with the purpose of this journal.

Again, thank you for giving us the opportunity to strengthen our manuscript with your valuable comments and queries. We have worked hard to incorporate your feedback and hope that these revisions persuade you to accept our submission.

Reviewer 3 Report

This study brings a huge benefit in terms of the body's reaction to metals. The harmful effect of Ni is known but this work adds to our knowledge of the action of this metal in the form of micro or nano or Ni ions when they are placed in living organisms such as mice. One area where the possibility of Ni allergies occurs is in dentistry where even orthodontic appliances for moving teeth or stainless steel used for tooth strengthening or drills used for grinding in the oral cavity, can trigger allergies, especially to Ni nanoparticles.

The work is complex and well-documented and the experimental part is logically conducted in order to obtain concrete data clearly proving the harmfulness, especially of Ni nanoparticles versus Ni microparticles. I recommend the paper for publication and congratulations to the team of authors.

Small corrections to the text are needed, such as:

Chapter 3.1.1 Please correct: 2796.0 ± 1471.7 nm instead of  2,796.0 ± 1471.7 nm

Figure 1. In the sentence ,,Scanning electron microscope images of (a) Ni- 244 NPs and (b) Ni-MPs are also shown” please correct c instead of a and d instead of b.

Chapter 3.1.2: 13.89 m2/g instead of m2/g

Author Response

Comments and Suggestions for Authors and Our Responses

This study brings a huge benefit in terms of the body's reaction to metals. The harmful effect of Ni is known but this work adds to our knowledge of the action of this metal in the form of micro or nano or Ni ions when they are placed in living organisms such as mice. One area where the possibility of Ni allergies occurs is in dentistry where even orthodontic appliances for moving teeth or stainless steel used for tooth strengthening or drills used for grinding in the oral cavity, can trigger allergies, especially to Ni nanoparticles.

The work is complex and well-documented and the experimental part is logically conducted in order to obtain concrete data clearly proving the harmfulness, especially of Ni nanoparticles versus Ni microparticles. I recommend the paper for publication and congratulations to the team of authors.

Small corrections to the text are needed, such as:

Chapter 3.1.1 Please correct: 2796.0 ± 1471.7 nm instead of  2,796.0 ± 1471.7 nm

Our response

As suggested by reviewer, the number has been corrected in current version of our manuscript (without comma).

Figure 1. In the sentence ,,Scanning electron microscope images of (a) Ni- 244 NPs and (b) Ni-MPs are also shown” please correct c instead of a and d instead of b.

Our response

As suggested by reviewer, the caption for Figure 1 has been corrected.

Chapter 3.1.2: 13.89 m2/g instead of m2/g

Our response

As suggested by reviewer, the text has been corrected.

Thank you for giving us the opportunity to strengthen our manuscript with your valuable comments and queries. We have worked hard to incorporate your feedback and hope that these revisions persuade you to accept our submission.